# Human-Relevant Sensitivity of iPSC-Derived Human Motor Neurons to BoNT/A1 and B1

**DOI:** 10.3390/toxins13080585

**Published:** 2021-08-22

**Authors:** Maren Schenke, Hélène-Christine Prause, Wiebke Bergforth, Adina Przykopanski, Andreas Rummel, Frank Klawonn, Bettina Seeger

**Affiliations:** 1Institute for Food Quality and Safety, Research Group Food Toxicology and Alternative/Complementary Methods to Animal Experiments, University of Veterinary Medicine Hannover, Foundation, 30173 Hannover, Germany; maren.schenke@tiho-hannover.de (M.S.); helene-christine.prause@univie.ac.at (H.-C.P.); wiebke.bergforth@tiho-hannover.de (W.B.); 2Department of Food Chemistry and Toxicology, Faculty of Chemistry, University of Vienna, 1090 Vienna, Austria; 3Institut für Toxikologie, Medizinische Hochschule Hannover (MHH), 30625 Hannover, Germany; przykopanski.adina@mh-hannover.de (A.P.); rummel.andreas@mh-hannover.de (A.R.); 4Biostatistics Research Group, Helmholtz Centre for Infection Research, 38124 Braunschweig, Germany; frank.klawonn@helmholtz-hzi.de; 5Department of Computer Science, Ostfalia University, 38302 Wolfenbüttel, Germany

**Keywords:** botulinum neurotoxins, motor neurons, alternative methods, in vitro potency determination

## Abstract

The application of botulinum neurotoxins (BoNTs) for medical treatments necessitates a potency quantification of these lethal bacterial toxins, resulting in the use of a large number of test animals. Available alternative methods are limited in their relevance, as they are based on rodent cells or neuroblastoma cell lines or applicable for single toxin serotypes only. Here, human motor neurons (MNs), which are the physiological target of BoNTs, were generated from induced pluripotent stem cells (iPSCs) and compared to the neuroblastoma cell line SiMa, which is often used in cell-based assays for BoNT potency determination. In comparison with the mouse bioassay, human MNs exhibit a superior sensitivity to the BoNT serotypes A1 and B1 at levels that are reflective of human sensitivity. SiMa cells were able to detect BoNT/A1, but with much lower sensitivity than human MNs and appear unsuitable to detect any BoNT/B1 activity. The MNs used for these experiments were generated according to three differentiation protocols, which resulted in distinct sensitivity levels. Molecular parameters such as receptor protein concentration and electrical activity of the MNs were analyzed, but are not predictive for BoNT sensitivity. These results show that human MNs from several sources should be considered in BoNT testing and that human MNs are a physiologically relevant model, which could be used to optimize current BoNT potency testing.

## 1. Introduction

Botulinum neurotoxins (BoNTs), produced by bacteria of the genus *Clostridium*, are the most lethal naturally occurring toxins, and yet they are used for medical applications. They exhibit a high selectivity for motor neurons (MNs) at the neuromuscular junction, where they specifically inhibit the release of neurotransmitters, inducing a potentially lethal muscle paralysis. This selectivity is employed when BoNTs are used to treat neuromuscular or autonomic disorders, pain conditions or for cosmetic applications [1]. The potency of a batch needs to be quantified prior to its release either by an ethically questionable mouse bioassay, which is the gold standard, or by alternative methods. However, despite the fact that the use of several alternative methods was enabled by the European pharmacopoeia in 2005, the number of animals used in batch testing appears to only have increased in the last decade, estimated to be as high as 400,000 mice annually in Europe alone [2]. Several factors impede the full replacement of the animal test, for instance, these in vitro assays are limited in their use to single products in addition to being proprietary while new products and manufacturers lacking validated alternatives are entering the European market [2,3,4,5,6]. For now, two BoNT serotypes, BoNT/A1 and B1 are routinely used, but more are undergoing clinical studies [7,8]. So far, seven major serotypes, A–G, with a multitude of subtypes, have been described, among which A, B, E and F can cause botulism in humans [9]. BoNT serotypes share a common pathway to enter cells and to convey their toxicity, but the key molecules involved differ and with that, different alternative methods may be required for potency determination. After injection, BoNTs bind with high specificity to the presynaptic membrane by adhering first to different complex polysialo gangliosides, such as GT1b, GD1a or GD1b, and then to different synaptic vesicle proteins, which are transiently exposed to the surface during synaptic cycling [10,11,12]. BoNT/A, D, E and F were shown to primarily bind to the Synaptic Vesicle Glycoprotein 2 (SV2) isoforms A, B and C, while BoNT/B and G bind to Synaptotagmin (SYT) isoforms 1 and 2 [13,14]. However, the affinity for a specific receptor isoform can vary between BoNT serotypes, as in the case of BoNT/B, for which SYT2 is the receptor with the highest affinity. As the mouse bioassay is the gold standard, BoNT activity is usually given in mouse lethality units. In humans and chimpanzees, however, a single site mutation occurred in the gene encoding SYT2 [15]. This resulted in a largely reduced affinity of BoNT/B to human SYT2, which mainly binds to the low affinity isoform SYT1 instead, causing a lower toxicity of BoNT/B in humans compared with mice. Thus, the dose of BoNT/B needs to be increased by a factor of 30–100 in comparison with BoNT/A for use in humans [16].

After attachment to a cell via their respective receptors, all BoNT serotypes are taken up into the cell via endocytosis. During the endosomal recycling process, the lumen of the vesicles is acidified, which induces a conformational change within the BoNT protein. The toxin consists of a light chain (LC) and a heavy chain (HC), the latter of which conveys the receptor binding and translocation of LC. The LC, a Zn^2+^-dependent endopeptidase, is released into the cytoplasm by the conformational change and cleaves specific target proteins, which are different soluble N-ethylmaleimide-sensitive-factor attachment receptor (SNARE) proteins such as the Synaptosomal-associated Protein of 25 kDa (SNAP25), the Vesicle-associated Membrane Protein (VAMP) and Syntaxin (STX) [1,17]. While the cleavage site is BoNT-serotype specific, as each serotype cleaves a single peptide-bond on one of the substrate proteins, the universal outcome is the inhibition of neurotransmitter release due to the loss of the essential function of SNARE proteins in exocytosis [18]. The number of steps involved in BoNT toxicity and the specificity of single serotypes for specific receptors and substrate proteins adds to the difficulty in replacing the mouse bioassay for the potency determination of BoNT-based pharmaceuticals [19]. Among available alternative methods are the ex vivo mouse phrenic nerve hemidiaphragm assay and a range of different types of in vitro assays, some of which are cell-based [20,21]. In many in vitro assays, the toxin or the cleaved substrate is detected with analytical or immunological methods, which omits one of the steps that dictates the sensitivity towards the toxin, which is the binding to a cell [22,23]. Cell-based assays, however, utilize different types of (neuroblastoma) cell lines, primary cells or neurons differentiated from stem cells, which can recapitulate all steps of toxin entry and activity [24,25]. The most physiologically relevant cell-based assay would be based on human MNs, as these can reflect all steps of toxicity relevant for BoNT pharmaceuticals and additionally should represent the sensitivity of humans. In a sense, BoNTs are tailored to specifically enter MNs, as these often exhibit high concentrations of receptor molecules that have a high affinity to BoNTs. The high-affinity isoforms SV2C and SYT2 are especially enriched in MNs and contribute to their sensitivity. The potential of human MNs to provide a more physiologically relevant and sensitive in vitro test system has been explored in several studies [26,27,28,29].

The aim of this study was to generate human MNs from induced pluripotent stem cells (iPSCs) with different protocols, to evaluate the sensitivity to BoNT/A1 and B1 and to compare it with that of the human neuroblastoma cell line SiMa, which has been used in several in vitro approaches [30,31,32,33]. By quantifying the cleavage of the respective substrates, we were able to show that human MNs can detect much lower activities of BoNT/A1 than the mouse bioassay and SiMa cells. In addition, human MNs could be used to quantify the potency of BoNT/B1, which SiMa cells were not capable of. The BoNT/B1 sensitivity of human MNs corresponded more with human sensitivity data than with the mouse bioassay, which indicates that these cells are a physiologically relevant model. Altogether, human MNs have a high potential to improve current alternative methods and to be used in universal methods that can detect more than one serotype at a time.

## 2. Results

### 2.1. Sensitivity of iPSC-Differentiated MNs and SiMa Cells to BoNT/A1 and B1

The application of BoNTs requires prior potency determination, for which only partial replacement of the mouse lethality test has been achieved [2]. In this study, we compare the sensitivity of MNs differentiated according to three different protocols to the sensitivity of the tumor cell line SiMa, which is used in several (proprietary) alternative methods [34]. To this end, SiMa cells were partially differentiated into a more neuronal phenotype and MN populations were generated as described in detail in Schenke et al. [28]. Human iPSCs (IMR90-04) were used for direct differentiation to MNs with protocols based on Du et al. [35] and Maury et al. [36], while the differentiation based on the publication by Kroehne et al. [37] is separated into differentiation of iPSCs to neuronal progenitor cells (NPCs), which can be expanded and differentiated further into MNs. The differentiated cells were treated with several concentrations of BoNT/A1 and B1 for 48 h and then analyzed for the cleavage of the respective BoNT substrate. The fraction of remaining full-length SNAP25 or VAMP2 after BoNT/A1 or B1 treatment, respectively, was analyzed via Western Blot, normalized to the total protein content per lane, and concentration–response data were modelled by nonlinear regression (Figure 1).

Both SiMa cells and MN populations showed a dose-dependent response to addition of BoNT/A1. The cells differentiated with the protocol based on Kroehne et al. [37] showed substrate cleavage at subpicomolar concentrations, with IC_50_ values of 0.0027 pM (77.6% confidence interval [CI]: 0.0004–0.0131 pM), followed by the protocol based on Maury et al. [36] with 0.0337 pM (77.6% CI: 0.0102–0.0818 pM) and the protocol based on Du et al. [35] 0.519 pM (77.6% CI: 0.014–1.979 pM). SNAP25 cleavage in SiMa cells occurred only at significantly higher concentrations, resulting in the IC_50_ value of 5.509 pM (77.6 % CI: 3.443–7.790 pM). When BoNT/B1 was used for the intoxication, detection of a dose–response correlation was only possible for MNs. In SiMa cells, the concentration of full-length VAMP2 varied in the analyzed replicates and showed no decrease, even at concentrations as high as 1000 pM BoNT/B1 (Figure 1A). When only untreated samples of SiMa cells were compared for full-length VAMP2, the concentration differed by a factor of four between the sample with the highest and lowest concentration, respectively (Figure 1B). Furthermore, a pronounced increase in the concentration of VAMP2 could be found in MNs after treatment with lower concentrations of BoNT/B1. In a small sample of blots probed with the antibodies for both substrate proteins, an increase in VAMP2 could also be seen when MNs generated according to the protocols based on Kroehne et al. [37] and Du et al. [35] were treated with BoNT/A1 (Figure A1, Appendix A). In MNs, higher concentrations of BoNT/B1 were necessary for the cleavage of VAMP2 compared with the cleavage of SNAP25 by BoNT/A1. IC_50_ values for BoNT/B1 treatment of cells differentiated with the protocols based on Maury et al. [36] were at 0.687 pM (77.6% CI: 0.319–1.560 pM), at 1.039 pM (77.6% CI: 0.766–3.314 pM) for the protocol based on Du et al. [35], which were significantly lower than the IC_50_ value of 14.84 pM (77.6% CI: 7.54–33.09 pM) for the protocol based on Kroehne et al. [37]. No IC_50_ value could be determined for SiMa cells treated with BoNT/B1.

From the IC_50_ values in pM, the correspondent mouse LD_50_/mL were calculated using mouse toxicity data generated by the BoNT manufacturer (Figure 2). As different batches of toxins were used, only the corresponding mouse LD_50_ units can be used for direct comparison of the protocols. Interestingly, cells generated by the protocol based on Maury et al. [36] were the most sensitive cell type for both BoNT/A1 and B1, with a sensitivity equating 0.05 and 4.12 mouse LD_50_/per mL. The cells differentiated according to the protocol based on Kroehne et al. [37] were only highly sensitive to BoNT/A1 with 0.06 mouse LD_50_/mL, while cells differentiated with the protocol based on Du et al. [35] showed intermediate sensitivity to BoNT/A1 and B1 with 10.90 and 6.24 mouse LD_50_/mL, respectively. SiMa cells were between 10 to 2000 times less sensitive to BoNT/A1 than the analyzed MN populations.

### 2.2. Molecular and Functional Differences between the Analyzed Cell Types

Large sensitivity differences among the MN populations generated with different protocols were detected regarding the BoNT serotypes A1 and B1. A factor that contributes to a high toxin affinity to MNs is the expression of corresponding protein receptors in high affinity isoforms. SiMa and MN populations generated with the differentiation protocols based on Du et al. [35], Kroehne et al. [37] and Maury et al. [36] were analyzed for their mRNA expression levels of proteins relevant for BoNT toxicity in Schenke et al. [28]. In this study, to account for the different protein turnover rates that might be encountered in mature neurons, receptors for BoNT/A1 and B1 were analyzed on the protein level by Western blot in MN populations, SiMa cells and compared with mouse brain synaptosomes (mSYSOs), which were used as a control (Figure 3).

All protein receptor isoforms except SV2B could be detected in MN populations and in SiMa cells, albeit at different concentrations. SV2A and SV2B were found in much higher concentrations in mSYSOs and were recorded with different exposure times. For SV2A, the highest concentration was detected in lysates from MNs generated based on the protocol by Du et al. [35], while the lowest concentration was found in MNs generated according to the protocol by Maury et al. [36]. The isoform SV2B could not be detected neither in MNs nor in SiMa cells. SV2C, however, which is found in high concentration in MNs, could be detected in higher concentration in the analyzed MN populations, but hardly in SiMa cells. The low-affinity receptor for BoNT/B, SYT1, was expressed in comparable concentrations in all analyzed lysates, while SYT2 was found only in MN and hardly in SiMa cells. For SYT1 and SYT2, the BoNT/B1 binding site was sequenced in IMR90, NPCs and SiMa cells and all sequences are consistent with the human genome (data not shown, alignment with RefSeq sequences NC_000012 [SYT1] and NG_041776 [SYT2]).

As the uptake of clostridial neurotoxins is dependent on the endocytosis, which mainly occurs during action potential-dependent exocytosis, we analyzed the electrical activity of the cultivated MNs on high-density microelectrode arrays (HD-MEAs). Progenitor cells from the protocols based on Du et al. [35], Kroehne et al. [37] and Maury et al. [36] were seeded on HD-MEAs at the latest time point prior to neurite outgrowth, which was at the stage of immature neurons for the protocol based on Du et al. [35] and at the MN progenitor stage for the other two protocols, and then differentiated further. The differentiations of the cells generated with the protocol based on Du et al. [35], Kroehne et al. [37] and Maury et al. [36] were finalized after 10, 15 and 21 days (days in vitro, DIV) on the HD-MEAs, respectively. The neuronal activity was recorded weekly after 7, 14 and 21 days on HD-MEAs, for which the entire chip (26,400 electrodes) was scanned for spiking activity (Figure 4).

The share of active electrodes was low (~1%) but mirrored the differentiation of the MN populations on the HD-MEAs, with the highest activity for each of the protocols being recorded close to the finalization of the differentiation. The mean spike amplitudes did not increase during the cultivation time and were below 40 µV, indicating immature neurons. Due to the low number of active electrodes, effects of BoNT intoxication could not be studied.

## 3. Discussion

### 3.1. Sensitivity of MNs to BoNT/A1 and B1 Is Superior to SiMa Cells

For potency determination of BoNT-based medicines only partial replacement of animal tests has been achieved [31,32,38,39]. Among potential test systems, human motor neurons (MNs) are the most physiologically relevant, being the cell type specifically targeted by BoNTs, and are able to recapitulate the major steps of toxin action [40]. In this study, MNs were generated from human iPSCs on the basis of three previously published differentiation protocols by Du et al. [35], Kroehne et al. [37] and Maury et al. [36]. MN populations generated with these protocols have been examined for their cellular composition, the presence of relevant ganglioside and protein receptors as well as of the SNARE substrates of all human relevant BoNT serotypes [28]. Here, the differentiated MNs as well as SiMa cells were used for treatment with BoNT/A1 and B1 and activity assessment via substrate cleavage quantification (Figure 1). For both serotypes, a dose-dependent decrease in full-length substrates could be seen in all MN populations. Initially, full-length VAMP2 increased in relation to the total protein loaded, after treatment with low concentrations of BoNT/B1. This is likely part of an adaptive mechanism, which might be explained by a generally reduced protein turnover, induced by the blockage of signal transmission or a specific upregulation of VAMP2, which has been described under certain circumstances, e.g., electric shock or chronic stress [41,42,43]. In SiMa cells, no dose-dependent decrease in VAMP2 was found even at high BoNT/B1 concentrations, but instead, the VAMP2 concentration showed high variation even in untreated samples. In other studies using anti-VAMP2 antibodies from other manufacturers, no VAMP2 was detected in SiMa cells [33,44]. Therefore, SiMa cells might be unsuited to assess VAMP2 cleavage by BoNT/B. From the dose–response data of MNs and SiMa cells treated with BoNT/A1 and MNs treated with BoNT/B1, different models of non-linear regression were used to calculate the IC_50_ values. As different batches of toxins were used in this study, these concentrations were translated into mouse LD_50_/mL by using mouse lethality data from the manufacturer. A reference volume of 1 mL was chosen, as a volume of 0.5 mL is usually injected during the mouse lethality assay, which results in a sensitivity of 2 mouse LD_50_/mL [25,45]. The sensitivity to BoNT/A1 was not only significantly higher (up to 2000 times) in the MN populations than in SiMa cells, but it was also at least 30 times higher than the mouse sensitivity. For MNs from commercial sources, a BoNT/A sensitivity with an IC_50_ of 0.39 pM was reported by Duchesne De Lamotte et al. [27] and of 0.12 mouse LD_50_/mL by Pellett et al. [26]. The MNs generated here according to the protocols by Kroehne et al. [37] and Maury et al. [36], which had yielded 14% and 16% Islet1-positive MNs in a previous study, showed sensitivities to BoNT/A1 of 0.0027 and 0.0337 pM, which correlates to 0.06 and 0.05 mouse LD_50_/mL. One of the major differences is the larger proportion of MNs in the commercial cells used by Duchesne De Lamotte et al. [27] (approximately 82%) and by Pellett et al. [26] (approximately 87%), but it seems worth investigating different neuronal populations, which might be more sensitive than pure MN populations.

For MNs generated based on the protocol by Du et al. [35], we have reported a sensitivity of 0.96 mouse LD_50_/mL from an initial experiment, which is lower than the results in this study by a factor of ten [28]. The same batch of toxin was used for both studies, which was stored at −80 °C as single use aliquots. However, a slightly adapted method for protein normalization was used based on whole protein concentration instead of using only actin for normalization, in addition to the use of a different software for quantification as well as a different model of nonlinear regression, which reduces the comparability. In our previous study cells were differentiated starting from the same progenitor batch, while here we started from iPSCs wanting to investigate the reproducibility after independent differentiations, which resulted in a larger confidence interval [28]. When establishing an assay based on MNs, it would be favorable to scale up batch production of progenitors for cryopreservation to maximize reproducibility.

For BoNT/B1, we detected sensitivities in MNs that were lower by a factor of 2 (Maury et al. [36] protocol) to 90 (Kroehne et al. [37] protocol) in comparison with the mouse lethality data, which has an LD_50_ of 2 units/mL. Considering the well-described interspecies sensitivity differences humans and mice towards BoNT/B by a factor of 30–100 due to the mutation of SYT2 in humans and chimpanzees, the lower sensitivity of human MNs reflects human sensitivity [12,15,16]. For the MN populations based on the protocols by Maury et al. [36] and Du et al. [35], we found a sensitivity of 4 and 6 mouse LD_50_/mL, which correlates well with 2 mouse LD_50_/mL reported by Pellett et al. [26]. However, according to the Western blot data, the sensitivity for single BoNT serotypes differs between the MN populations used in this study with the cells differentiated with the protocol based on Du et al. [35] being less sensitive to BoNT/A1, and with the cells differentiated with the protocol based on Kroehne et al. [37] showing low sensitivity to BoNT/B1. In Schenke et al. [28], we have reported a MN yield (Islet-positive cells) of 51%, 16% and 14% for the protocols based on Du et al. [35], Maury et al. [36] and Kroehne et al. [37], respectively. Only the cells differentiated according to Maury et al. [36] detected low concentrations of both BoNT/A1 and B1 (0.05 and 4.12 mouse LD_50_/mL), even though the percentage of MNs was lower than in the cells differentiated with the protocol based on Du et al. [35], indicating that MN percentage is not the only factor determining the sensitivity.

### 3.2. Factors Contributing to MN Sensitivity

Among the factors that determine the BoNT sensitivity of a cell, which might be synergistic, are the expression and distribution of isoforms of substrates, protein and ganglioside receptors, glycosylation of receptors, cell maturity and synaptic activity [18,26,46,47]. Factors such as the glycosylation of receptors have been discussed as a reason for individual differences in patients treated with BoNTs [1,18]. The gene expression of protein receptors and substrates were previously assessed in Schenke et al. [28] and could be detected in MN populations generated with each of the protocols. As the protein turnover in mature neurons might be decreased, the concentration of the BoNT receptors was assessed via Western blot (Figure 3). However, the protein receptor concentrations largely mirror the gene expression data in Schenke et al. [28] with deviations with respect to SV2B, which is 10-fold less expressed than SV2A in all MNs and SiMA cells, but was not detected in any MNs or SiMa cells by Western blot. Additionally, SV2C and SYT2 are hardly detectable on the protein level in SiMa cells, whereas RT-qPCR yielded comparable expression level between MNs and SiMa cells for SYT2 and only slightly smaller SV2C expression in SiMa cells compared with MNs. In comparison with mSYSOs, it appears that the concentration of SYT2, which is the dominant isoform in MNs, was lower than the concentration of SYT1 in the analyzed MNs [48]. Since the share of MNs in these populations is only 14–51% on average, other cell types with other isoform expression patterns can have affected the average protein levels.

Both the gene expression data and the receptor protein concentration cannot explain the sensitivity differences alone, as, e.g., the cells differentiated according to the protocol based on Kroehne et al. [37] exhibit a relatively high concentration of both SYT1 and SYT2, but low BoNT/B1 sensitivity (Figure 2). Additionally, the cells used for differentiation (IMR90-04 and NPCs) as well as SiMa cells showed full accordance in the DNA sequence encoding the BoNT/B binding site in SYT1 and SYT2 of the human genome. Since BoNT/B has stronger autonomic side effects than BoNT/A, it is expected to enter other subsets of neurons, which might be a reason for serotype differences seen in this study [16].

As another parameter to affect BoNT sensitivity, the neuronal maturation and synaptic activity of MN populations were studied with MEAs. For this purpose, MN progenitor cells, not yet showing neurite outgrowth, from the different protocols were seeded and differentiated further on HD-MEA chips. Therefore, the cultivation time on the MEAs is different from the differentiation day. Cells from the protocols based on Du et al. [35], Kroehne et al. [37] and Maury et al. [36] reached maturation after 10, 15 and 21 days on the MEAs, respectively. In the data recorded here, neuronal activity was low and did not increase after finalization of the differentiation on the day set by the authors of the original differentiation protocols. In addition, the low spike amplitudes resemble rather immature neurons [49]. Due to this, the activity of MNs needs to be improved prior to further experiments such as BoNT potency evaluation utilizing MEAs. The low neuronal activity, especially in the MN populations generated with the protocols by Kroehne et al. [37] and Du et al. [35], might be due to experimental parameters such as evaporation, which was substantial in MEA wells. The activity might be optimized by adapting the cultivation parameters by, e.g., using different cell culture media or co-culture of MNs with astrocytes or Schwann cells. Co-culture with astrocytes has been shown to shorten the time needed until functional maturity is reached and to improve the neuronal activity [50]. A slightly greater functional maturity from the cells generated with the protocol based on Maury et al. [36] compared with the other protocols is apparent from the recorded neuronal activity and may have contributed to the superior sensitivity of these cells. Optimizing the neuronal activity might further increase the sensitivity of MNs to different BoNTs and while studying BoNT intoxication of MNs on MEAs is challenging, it remains a promising system.

The factors not analyzed in this study, such as the intra- and intercellular distribution of the relevant proteins, their glycosylation and the presence of gangliosides, might have played a role in the apparent sensitivity of the MN populations. As neuronal differentiations in vitro can yield heterogeneous cell compositions, these factors may differ depending on the exact composition. For instance, the binding affinity of BoNT/B to SYT2 is largely enhanced by the presence of gangliosides [40,51,52,53]. Therefore, cells with, e.g., a high expression of SYT1/2, which lack gangliosides, will have a low sensitivity to BoNT/B, DC and the G. A method suited for analyzing single cells for several parameters, such as the expression of multiple proteins, which is flow cytometry. This method is rarely used for neurons, and with the cells generated in this study, the strong interconnectivity of MNs with their particularly long axons made the generation of a viable single cell suspension virtually impossible. Ultimately, however, it will be necessary to look at the endpoint of BoNT toxicity, the inhibition of neurotransmitter release. The multitude of factors that contribute to the sensitivity of a given cell, e.g., the distribution of BoNT substrates within a cell, could also affect substrate cleavage and the inhibition of exocytosis [47]. Therefore, to have a BoNT in vitro assay, that can faithfully represent human sensitivity, either in already available or new pharmacological preparations from different BoNT serotypes or subtypes, it should be based on quantification of neurotransmission inhibition.

The data shown here aim to emphasize that human MNs are suitable to be used in further optimization of alternative methods for BoNT potency determination, likely in combination with MEAs or reporters of neurotransmitter release [54,55]. Since the sensitivity of a cell population could not be predicted with single parameters alone, it appears necessary to further investigate the potential of (motor) neurons from different sources. Nevertheless, the MNs generated in this study, depending on the protocol used, were able to quantify pico- to subpicomolar concentrations of both BoNT/A1 and B1, were significantly more sensitive than the tumor cell line SiMa for BoNT/A1 and could reflect human sensitivity, all of which makes human MNs a potential foundation for serotype-independent BoNT potency determination.

## 4. Materials and Methods

### 4.1. Cell Culture

All cells were cultivated at 37 °C, 5% CO_2_, 95% rel. humidity in an incubator and checked for Mycoplasma contamination monthly. MNs were differentiated from the human iPSC line IMR90-04 (IMR90), which was originally purchased from WiCell [56]. IMR90 were cultivated in StemMACS™ iPS-Brew XF medium (Miltenyi, Bergisch Gladbach, Germany) with medium changed every other day on plates that were coated with Corning^®^ Matrigel^®^ basement membrane preparation (growth factor reduced; #354230, Corning, New York, NY, USA). At 70–80% confluency, IMR90 were passaged with 0.02% EDTA in phosphate-buffered saline without Mg^2+^ and Ca^2+^ (PBS). To increase the survival of the cells, 10 µM Rho-Kinase inhibitor Y-27632 (TargetMol, Boston, USA) was supplemented after thawing and subculturing of IMR90. For neuron differentiation, neural medium with differentiation-stage specific supplements was used. Neural medium consisted of equal volumes of DMEM/F12 (#21331046, Thermo Fisher Scientific, Waltham, Massachusetts) and MACS^®^ Neuro medium (Miltenyi), as well as 0.5× N-2 Supplement (Thermo Fisher Scientific), 1× MACS^®^ NeuroBrew^®^-21 (Miltenyi), 1× L-glutamine (Thermo Fisher Scientific) and 1× penicillin/streptomycin (P/S, Thermo Fisher Scientific). The protocols used for the differentiation are described in detail in Schenke et al. [28], with a brief summary given below.

### 4.2. Differentiation Based on Kroehne et al. 

One of the MN differentiation protocols used in this study is based on the publication of Kroehne et al. [37]. It consists of two steps, which are the differentiation of a proliferative population of neuronal progenitor cells (NPCs) from iPSCs and the following differentiation into a neuronal population that contains MNs. Briefly, IMR90 were cultivated on mitotically inactive mouse embryonic fibroblasts (MEFs, Cell Biolabs Inc., Heidelberg, Germany), detached mechanically and enzymatically and cultivated in suspension in the form of embryoid bodies. Neuralization of embryoid bodies was induced with 1 µM Dorsomorphin (DM, Abcam, Berlin, Germany), 3 µM CHIR99021 (CHIR, Axon Medchem, Groningen, Netherlands), 0.5 µM Purmorphamine (PMA, Stemcell, Cologne, Germany) and 10 µM SB431542 (SB, MedChemExpress, Monmouth Junction, NJ, USA) for 4 days, followed by cultivation in 3 µM CHIR, 0.5 µM PMA and 150 µM ascorbic acid (AA, Sigma-Aldrich, St. Louis, MI, USA). The spheres were dissociated mechanically after 6 days in total and seeded on Matrigel-coated plates. After passaging with Accutase (Sigma-Aldrich) four times, neural medium with 0.5 µM SAG (MedChemExpress) instead of PMA was used. NPCs were Nestin and SOX1 positive. For differentiation into motor neurons, NPC cultures that have been passaged at least 13 times were detached with Accutase and seeded at 2 × 10^5^ cells/mL on Matrigel-coated plates in neural medium supplemented with 200 µM AA, 0.5 µM SAG, 1 µM retinoic acid (RA, Sigma-Aldrich), 1 ng/mL GDNF and 2 ng/mL BDNF (both from Peprotech, Hamburg, Germany). At day 6, cells were replated and cultivated in neural medium containing 200 µM AA, 2 ng/mL GDNF and BDNF, 1 ng/mL TGFß3 (Sigma-Aldrich), 200 µM dbcAMP (MedChemExpress) and 10 µM DAPT (MedChemExpress) until mature neurons were obtained on day 21.

### 4.3. Differentiation Based on Du et al.

For the MN differentiation based on the protocol by Du et al. [35], IMR90 were seeded on Matrigel-coated plates at 5 × 10^4^ cells per mL in StemMACS™ iPS-Brew XF medium with 10 µM Y-27632. The following day, neural medium was used, which was then exchanged every other day. For neuronal induction, 3 µM CHIR, 2 µM DMH1 (Bertin Pharma, Montigny le Bretonneux, France) and 2 µM SB were used from day 1 to day 6. Cells were passaged and cultivated in neural medium with 0.1 µM RA, 0.5 µM PMA, 1 µM CHIR, 2 µM DMH1 and 2 µM SB for six more days. Motor neuron progenitors were detached with Collagenase and transferred in neural medium with 0.5 µM RA and 0.1 µM PMA to low attachment plates (Corning), where they formed neurospheres. After 6 days, the spheres were dissociated, seeded on Matrigel coated plates and neurons maturated until day 28 in neural medium containing 0.5 µM RA, 0.1 µM PMA, 0.1 µM compound E (CE, Bertin Pharma), 2 ng/mL of GDNF, BDNF and CNTF (Peprotech).

### 4.4. Differentiation Based on Maury et al.

The MN differentiation based on the publication by Maury et al. [36] was conducted as follows: IMR90 were seeded as small aggregates with a density of 1 × 10^5^ cells/mL in low-attachment plates in neural medium containing 0.5 µM AA, 3 µM CHIR, 2 µM DMH1, 2 µM SB and 5 µM Y-27632 and cultivated in suspension. Totals of 0.1 µM RA and 0.5 µM SAG were added and Y-27632 was withdrawn on day 2. On day 4, only AA, RA and SAG were added until motor neuron progenitors were obtained on day 9, where neurospheres were dissociated mechanically and enzymatically and seeded on Matrigel-coated plates. From day 9 to 14, 10 µM DAPT and from day 11 to 32, GDNF, BDNF and CNTF (5 ng/mL) and 1 µg/mL dbcAMP were added.

### 4.5. Cultivation and Differentiation of SiMa Cells

The neuroblastoma cell line SiMa (ACC 164), which was kindly provided by G. Püschel, was cultivated in RPMI1640 (Biochrom, Cambridge, UK) with 10% inactivated FCS (Biochrom), 1% P/S and 1% L-glutamine. SiMa cells were differentiated into a more neuronal phenotype by replacing the serum with 2% MACS^®^ NeuroBrew^®^-21, 1% N-2 Supplement and 1 mM non-essential amino acids. SIMA were differentiated on poly-L-lysine (Sigma)-coated plates for two days before BoNTs were added.

### 4.6. Analysis of SNAP25 and VAMP2 Cleavage

Cells differentiated and maturated according to the protocols based on Kroehne et al. [37], Du et al. [35] and Maury et al. [36] were treated with purified BoNT/A1 or B1 (Miprolab, Göttingen, Germany; #3101 and #3201) diluted in neural medium. BoNT/A1 from one batch (2.8 × 10^8^ minimal lethal doses/mg) was used for treatment of all cells except for the cells differentiated according to Maury et al. [36], for which another batch (2 × 10^7^ minimal lethal doses/mg) was used. BoNT/B1 from one batch (1.6 × 10^8^ minimal lethal doses/mg) was used for treatment of all cells except for the cells differentiated according to Du et al. [35] and Maury et al. [36], for which another batch (8 × 10^7^ minimal lethal doses/mg) was used. For the treatment of SiMa cells, toxins were diluted in SiMa differentiation medium, while neural medium without supplements was used for treatment of MNs. After 48 h, cells were washed with PBS, harvested in RIPA buffer (50 mM Tris, 150 mM NaCl, 1 mM EDTA, 1% Triton X100, 1% sodium deoxycholate, 0.1% SDS, 1% Protease Inhibitor Cocktail Set III) and lysed with ultrasonication. The cleared lysate (25 µg SiMa Lysate, 10 µg MN lysate) was separated with Any kD™ Mini-PROTEAN^®^ TGX™ gels (Bio-Rad, Feldkirchen, Germany) and transferred onto nitrocellulose membranes (GE Healthcare, Freiburg, Germany). Membranes were blocked with 5% milk powder in TBST for 1 h. Anti-SNAP25 and anti VAMP2 primary antibodies (Table A1) were diluted 1:1000 in 5% milk powder in TBST and incubated over night at 4 °C. Horseradish peroxidase (HRP) coupled secondary antibody (Sigma, Table A2) was diluted 1:20,000 and incubated for 1 h at room temperature. The antibodies directed against SNAP25 and VAMP2 recognize only the respective full-length proteins as they bind to the carboxy terminus, which is cleaved off by BoNT/A1 and B1. Bands were visualized with Immobilon Western HRP Substrate (Merck/Millipore, Burlington, MA, USA) and quantified with the ChemiDoc imaging system (Bio-Rad). Afterwards, the membrane was stained with India Ink (Pelikan #4001) with 2% acetic acid for 30 min, washed with H_2_O, quantified with colorimetric detection and used to normalize the signal intensities to the total protein loaded per band. Toxin-treated samples were normalized to the respective untreated control. Concentration–response data were modelled by nonlinear regression as described in Seeger et al. [57] in RStudio (RStudio Team 2021, PBC, Boston, MA, USA, http://www.rstudio.com/, accessed on 19 August 2021). The used regression models were chosen according to the Akaike Information Criterion (AIC) from probit, logit, Weibull, Aranda-Ordaz, generalized logit I and II. Confidence intervals (CIs) were estimated via 1000 bootstrap simulations. IC_50_ values were computed as the point where the corresponding curve crosses the midpoint of range (codamain) of the corresponding approximation function. Since standard hypothesis tests cannot be applied directly to check whether the difference in IC_50_ values might be explainable by random effects, CIs were used. The probability for (1-α) CIs to overlap under the null hypothesis that the data originate from the same distribution is α^2^. To obtain the usual significance level of 5% for hypothesis tests, $\alpha =\sqrt{0.05}=0.224$ needs to be chosen, i.e., 77.6% CIs were considered. Non-overlapping CIs indicate significantly different IC_50_ values.

### 4.7. Analysis of BoNT Receptors

Mouse brain synaptosomes were generated as detailed by Rummel et al. [58]. Briefly, functional synaptosomes were recovered from a Percoll gradient after homogenisation of cerebrum obtained from 6–10 mouse brains and various centrifugation steps, and finally diluted in physiological buffer (140 mM NaCl, 5 mM KCl, 1 mM MgCl_2_, 1 mM CaCl_2_, 20 mM HEPES, 10 mM glucose, 0.5% BSA, pH 7.4) with the final synaptosomal protein adjusted to a concentration of 10 mg/mL. Lysates of MNs and SiMa cells not treated with BoNTs were pooled from three independent differentiations. Totals of 10 µg protein each were additionally separated on a 12.5% gel via SDS-PAGE and transferred onto PVDF membranes (Roti^®^-PVDF, pore size 0.45 µm, ROTH, Karlsruhe, Germany). Membranes were blocked with 5% milk powder in PBST (PBS, 0.05% Tween) for 1 h. Anti-SV2A, anti-SV2B, anti-SV2C, anti-SYT1 and anti-SYT2 primary antibodies (Table A1) were diluted in 5% milk powder in PBST and incubated over night at 4 °C. HRP-coupled secondary antibodies (both from Rockland, Limerick, PA, USA, Table A2) were diluted 1:20,000 and incubated for 4 h at room temperature. Bands were visualized with SuperSignal West Femto Maximum Sensitivity Substrate (Thermo Scientific) with the ChemoStar ECL Imager (Intas, Göttingen, Germany).

### 4.8. Microelectrode Arrays

For recording of electrical activity of MNs, MN progenitors from the protocols based on Du et al. [35], Kroehne et al. [37] and Maury et al. [36] were each seeded on six MaxOne high-density (HD) MEAs (MaxWell Biosystems, Zurich, Switzerland). MaxOne HD-MEAs consist of a single well with 26,400 electrodes, of which 1024 can be recorded simultaneously on a sensor area of 3.85 × 2.10 mm^2^, with a center-to-center electrode distance of 17.5 µm, which facilitates subcellular resolution. To enhance the attachment of the neurons to the surface, HD-MEAs were prepared as recommended by the manufacturer: At first, HD-MEAs were pretreated with 1% Terg-a-zyme^®^ enzyme detergent (Sigma) over night at room temperature. On the next day, HD-MEAs were thoroughly rinsed with H_2_O, sterilized with 70% Ethanol in H_2_O, dried, placed in 100 mm Petri dishes and coated with 50 µL of 0.1 mg/mL poly-D-lysine (Thermo Fisher Scientific) for 1 h at 37 °C. HD-MEAs were rinsed three times with H_2_O, allowed to dry for 30 min and then coated with 50 µL of Geltrex™ hESC-Qualified, Ready-To-Use, Reduced Growth Factor Basement Membrane Matrix (Thermo Fisher Scientific). 30 mm dishes filled with H_2_O were placed next to the HD-MEAs to reduce evaporation. After incubation for 1 h at 37 °C, Geltrex was removed, and MN progenitors or immature neurons were seeded on the HD-MEAs in the respective differentiation medium. As the differentiation protocols differed with regard to the timing of subcultivation and choice of suspension or adherent culture, the latest time point where cells were subcultivated prior to neurite formation was chosen for plating of HD-MEAs. Additionally, the number of cells seeded was adjusted to ensure similar cell density on the MEAs and to compensate for cell death after thawing. For the protocol based on Du et al. [35], 150,000 immature neurons frozen on differentiation day 18, were seeded onto the sensor and surrounding area in a volume of 50 µL. For the protocols based on Kroehne et al. [37] and Maury et al. [36], 150,000 and 450,000 MN progenitors frozen on differentiation day 6 and day 9, respectively, were seeded onto the sensor and surrounding area in a volume of 50 µL. After cells were allowed to attach for 1 h in the incubator, 600 µL of respective differentiation medium was added. In parallel to the MEAs, cells were seeded on similarly coated cover slips at the same density in relation to the surface area, in a volume of 500 µL. A total of 50% of the cultivation medium was exchanged as required by the respective differentiation protocol until the differentiation was finalized, after which at least 50% of the medium was exchanged twice a week. For the differentiation based on Maury et al. [36], 10 µM DAPT was supplemented during the whole cultivation time to prevent overgrowth of neuronal progenitors, which was only added until differentiation day 14 in the other experiments with this protocol. Electrical activity and network properties of neurons on HD-MEAs were assessed weekly with a MaxOne recording unit (MaxWell Biosystems), which was placed inside the incubator, starting after seven days of cultivation on the chips. To distinguish from the differentiation day, days in vitro (DIV) is used to describe the cultivation time on the MEAs. Spiking activity on the whole sensor area was assessed with the “activity scan” module of the MaxLab Live software (MaxWell Biosystems) in sequential configurations, each of which was recorded for 30 s. Recorded metrics are the spike amplitude (90th percentile of the negative amplitude of detected spikes in µV) and the active area (electrodes with firing rate > 0.1 Hz and spike amplitude > 20 μV).

## Figures and Tables

**Figure 1 toxins-13-00585-f001:**
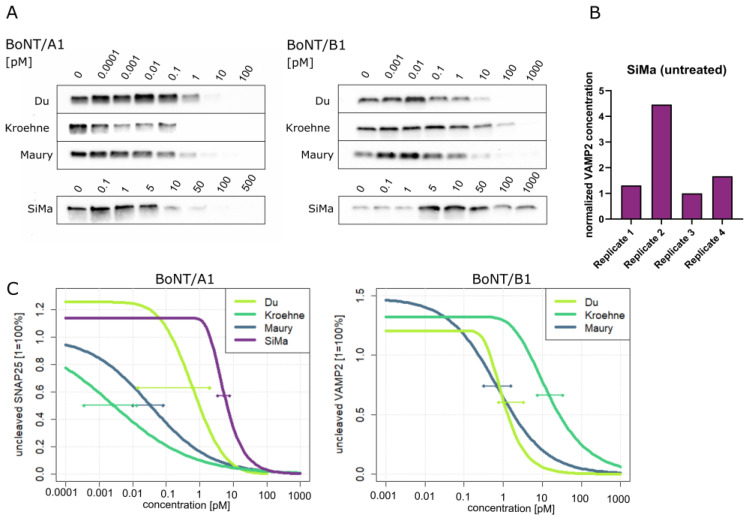
Cleavage of the respective substrates SNAP25 or VAMP2 by BoNT/A1 and B1. (**A**) Representative Western blots with detection of full-length substrate protein in motor neurons (MNs, n = 3) generated with the protocols based on Du et al. [35], Kroehne et al. [37] and Maury et al. [36], as well as SiMa cells treated with different concentrations of BoNT/A1 and B1 (SiMa cells: n = 3 for BoNT/A1, n = 4 for BoNT/B1). (**B**) VAMP2 concentration in untreated SiMa cells normalized to the total protein per lane and to the replicate with the lowest intensity. (**C**) Dose–response curves of normalized substrate concentration for MNs and SiMa cells treated with BoNT/A1 and MNs treated with BoNT/B1 were modelled with nonlinear regression. IC_50_ values for the respective curves are indicated with their 77.6% confidence intervals. To obtain the significance level of 5% for hypothesis tests, $\alpha =\sqrt{0.05}=0.224$ was chosen and 77.6% CIs considered.

**Figure 2 toxins-13-00585-f002:**
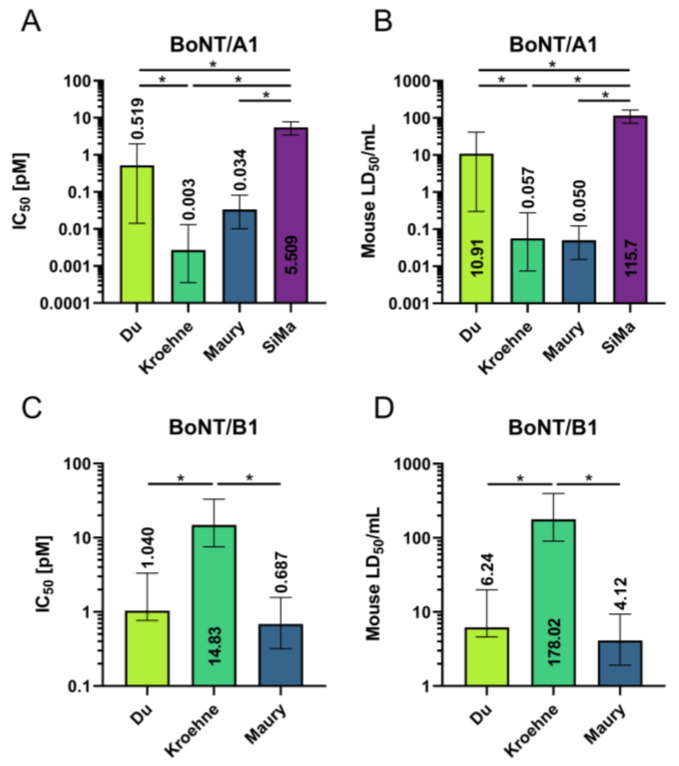
Sensitivity of MN populations differentiated according to the protocols by Du et al. [35], Kroehne et al. [37] and Maury et al. [36] as well as SiMa cells (all with n = 3) to BoNT/A1 and B1 in IC_50_ [pM] (**A**,**C**) and mouse LD_50_/mL (**B**,**D**). Data provided from the manufacturer were used to estimate mouse LD_50_/mL, with details regarding the batches used given in Section 4.6. Mean IC_50_ values or mouse LD_50_/mL are shown with their respective 77.6% confidence intervals. To obtain the significance level of 5% for hypothesis tests $\alpha =\sqrt{0.05}=0.224$ was chosen, and 77.6% CIs considered. Non-overlapping CIs show significantly different IC_50_ values indicated by *.

**Figure 3 toxins-13-00585-f003:**
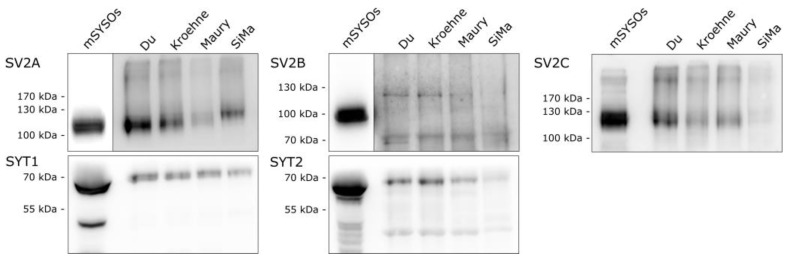
Expression of synaptic vesicle proteins SV2A/B/C and SYT1/2 in MNs generated according to the protocols by Du et al. [35], Kroehne et al. [37] and Maury et al. [36] as well as SiMa cells were detected via Western blot using isoform-specific antibodies for detection. Each panel depicts the detection with one of the antibodies. A total of 10 µg of lysate pooled from three independent differentiations was analyzed and compared with mouse brain synaptosomes (mSYSOs). For SV2A and SV2B blots, the signal intensity from mSYSOs was far greater than for the other samples, and images with different exposure times were combined (Exposure time of cell lysates: SV2A 1 min, SV2B 10 min and for mSYSOs 2 s and 10 s, respectively).

**Figure 4 toxins-13-00585-f004:**
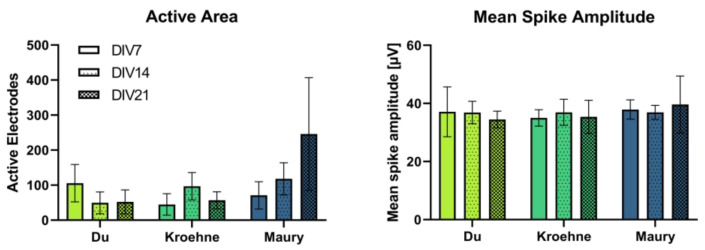
Activity of MN populations differentiated according to the protocols by Du et al. [35], Kroehne et al. [37] and Maury et al. [36] on high-density microelectrode arrays (HD-MEAs) after 7, 14 and 21 days on the chips (days in vitro, DIV). Cells from one differentiation each were seeded on six HD-MEAs, respectively. Recorded parameter are the active area and the mean spike amplitude. Data represented as mean ± standard deviation, n = 6.

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
