# Peer review of "Human-Relevant Sensitivity of iPSC-Derived Human Motor Neurons to BoNT/A1 and B1"

_toxins, 2021, doi:10.3390/toxins13080585_

Round 1
Reviewer 1 Report
In the paper, the authors aimed to generate human MNs from induced pluripotent stem
cells (iPSCs) with different protocols and to evaluate the sensitivity to BoNT/A1 and B1
in order to compare it to the SiMa. The authors were able to show human MNs' ability to detect lower activities of BoNT/A1 than the mouse bioassay and SiMa cells.
In addition, the authors showed that human MNs could be used to quantify the potency of BoNT/B1, which SiMa cells were not capable of.
The paper is well-written, it is well-supported by the literature and methods/results are presented with sufficient information. I suggest accepting the paper after minor suggestion.
Minor suggestion:
1) Table A1 - under Dilution column 1.1000 should be 1:1000?
Reviewer 2 Report
The aim is the determination of the sensitivity of iPSC-differentiated human motor neurons to 2 BoNT/A1 and B1, and found that the human motor neuron is superior to the established neuroblastoma 3 cell line SiMa. The authors describe the technique with precision. The methodology is appropriate. This article gives very interesting elements of understanding the direct iPSC differentiated human motor neurons could be standardized tool for avoiding animal bioassays. Partial improvement of the text would help readers of “Toxins” understanding the article.
Title should be amended that it reveals the conclusion of the study, and the title does not cover all of your studies in one sentence. I would recommend revising it from “Sensitivity of iPSC-differentiated human motor neurons to 2 BoNT/A1 and B1 is superior to the established neuroblastoma 3 cell line SiMa” to “Sensitivity of iPSC-differentiated human motor neurons in 2 BoNT/A1 and B1”.
Abstract: Please have “iPSC” in the abstract not to be abbreviated form. It would be better to have cosmetics deleted in the abstract section. Cosmetics are one of the medical treatments that is unnecessary in the abstract.
Introduction: I would shorten introduction by focusing more on serotypes A and B, which is in clinical use of. I would specify serotype A and B. “For now, two BoNT serotypes (A and B) are routinely used, but more are undergoing clinical studies [7,8].” Is the following paragraph necessary? “So far, seven major serotypes, A-G, with a multitude of 43 subtypes, have been described. Among them, A, B, E and F can cause Botulism in humans 44 with lethal oral doses as low as 1 µg/kg body weight [9]..”
Result: It is well written with detailed, understandable data presentation with the figures, which are adequate.
Discussion and method: It is a well designed study investigating the use of iPSC-differentiated human motor neurons. However, the discussion are too long with unnecessary background information that are out of points. I would recommend shortening the discussion and having a paragraph of conclusion at the last of the discussion.
